# Inflammasome-Mediated Immunogenicity of Clinical and Experimental Vaccine Adjuvants

**DOI:** 10.3390/vaccines8030554

**Published:** 2020-09-22

**Authors:** Sören Reinke, Aneesh Thakur, Cillian Gartlan, Jelena S. Bezbradica, Anita Milicic

**Affiliations:** 1The Jenner Institute, Nuffield Department of Medicine, University of Oxford, Oxford OX3 7DQ, UK; 2Department of Pharmacy, Faculty of Health and Medical Sciences, University of Copenhagen, 2100 Copenhagen, Denmark; aneesh.thakur@sund.ku.dk; 3Trinity Biomedical Sciences Institute, School of Biochemistry and Immunology, Trinity College Dublin, Dublin D02 R590, Ireland; gartlanc@tcd.ie; 4Kennedy Institute of Rheumatology Research, Nuffield Department of Orthopaedics, Rheumatology and Musculoskeletal Sciences, Medical Sciences Division, University of Oxford, Oxford OX3 7FY, UK; jelena.bezbradica@kennedy.ox.ac.uk

**Keywords:** inflammasome, adjuvant, vaccine, immunogenicity, NLRP3, NLRC4, AIM2, pyrin, non-canonical, caspase-1

## Abstract

In modern vaccines, adjuvants can be sophisticated immunological tools to promote robust and long-lasting protection against prevalent diseases. However, there is an urgent need to improve immunogenicity of vaccines in order to protect mankind from life-threatening diseases such as AIDS, malaria or, most recently, COVID-19. Therefore, it is important to understand the cellular and molecular mechanisms of action of vaccine adjuvants, which generally trigger the innate immune system to enhance signal transition to adaptive immunity, resulting in pathogen-specific protection. Thus, improved understanding of vaccine adjuvant mechanisms may aid in the design of “intelligent” vaccines to provide robust protection from pathogens. Various commonly used clinical adjuvants, such as aluminium salts, saponins or emulsions, have been identified as activators of inflammasomes - multiprotein signalling platforms that drive activation of inflammatory caspases, resulting in secretion of pro-inflammatory cytokines of the IL-1 family. Importantly, these cytokines affect the cellular and humoral arms of adaptive immunity, which indicates that inflammasomes represent a valuable target of vaccine adjuvants. In this review, we highlight the impact of different inflammasomes on vaccine adjuvant-induced immune responses regarding their mechanisms and immunogenicity. In this context, we focus on clinically relevant adjuvants that have been shown to activate the NLRP3 inflammasome and also present various experimental adjuvants that activate the NLRP3-, NLRC4-, AIM2-, pyrin-, or non-canonical inflammasomes and could have the potential to improve future vaccines. Together, we provide a comprehensive overview on vaccine adjuvants that are known, or suggested, to promote immunogenicity through inflammasome-mediated signalling.

## 1. Introduction

In the development of modern vaccines, adjuvants play a pivotal role as they can enhance protective immunity and usually provide an improved safety profile in comparison to live attenuated vaccines [1]. As early as 1924, Gaston Ramon described the first vaccine adjuvants, which enhanced the immune response to diphtheria and tetanus vaccines [2]. In addition to augmenting the immune response in general, adjuvants can also allow vaccine dose sparing - which would enable to increase global vaccine supply, reduce the number of immunisations, enhance vaccine efficacy in immuno-compromised individuals, such as young children with a developing immune system and the elderly, or broaden the immune response against highly variable pathogens, e.g., influenza [3]. Of note, vaccine adjuvant design and choice should always specifically address the targeted pathogen in order to activate the appropriate specific pathways. Thus, adjuvants qualitatively and quantitatively direct the immune system to initiate a pathogen-specific response. Today, a broad range of compounds such as mineral salts, water and oil emulsions, saponins, liposomes, microparticles and pattern recognition receptors (PRR)/Toll-like receptors (TLR) agonists are known for their adjuvanticity [4]. Although different adjuvants have diverse modes of action (MoA) in promoting immunity (reviewed in [5]), the common feature is to foster activation of innate immune cells, such as dendritic cells (DCs), monocytes, macrophages or neutrophils in order to boost activation of T- and B-cells, which are then able to mediate robust and long-lasting immunity against a specific pathogen. In this context, adjuvants can primarily perform one or more of several functions [5]: (I) creating a depot effect to maintain the release of antigen at the site of injection, (II) boosting the secretion of cytokines and chemokines, (III) enhancing the recruitment of innate immune cells at the site of injection, (IV) stimulating antigen uptake by antigen presenting cells (APCs), (V) enhancing APC maturation/expression of major histocompatibility complex (MHC) class II and co-stimulatory molecules and migration to draining lymph nodes (dLN) and, importantly, (VI) activating the inflammasome, which is the main focus of this review. An overview of inflammasome-activating adjuvants can be found in Table 1.

Inflammasomes are cytosolic multiprotein signalling platforms that drive the activation of inflammatory caspases [54]. ‘Canonical’ inflammasome complexes are made of specialised (germline-encoded) PRRs that couple to the effector enzyme pro-caspase-1, typically via adaptor molecule called apoptosis speck-like protein containing a caspase activation and recruitment domain (ASC) [55]. Best characterised PRRs that make up ‘canonical inflammasomes’ are nucleotide-binding domain, leucine-rich repeat receptor (NLR) family, pyrin domain containing 1 (NLRP1); NLR family, pyrin domain containing 3 (NLRP3); NLR family, caspase activation and recruitment domain containing 4 (NLRC4); absent in melanoma 2 (AIM2), and pyrin. There is also a ‘non-canonical’ inflammasome, made of inflammatory caspase-11 in mice, and caspase-4 or -5 in humans [55]. In general, inflammasomes recognise diverse pathogen-associated molecular patterns (PAMPs), danger-associated molecular patterns (DAMPs), or the loss of cellular homeostasis. All canonical inflammasomes drive a common downstream response: activation of pro-caspase-1 to cleave pro-inflammatory cytokines interleukin-1 beta (IL-1β) and IL-18 and induce the secretion of their active forms. Activated caspase-1 also cleaves a pore-forming protein Gasdermin D (GSDMD), to create a large plasma membrane pore in the cell in which the inflammasome was activated. GSDMD pore allows rapid release of IL-1β and IL-18, causes cell swelling and ultimately leads to a pro-inflammatory form of cell death, called pyroptosis. Pyroptosis releases more DAMPs and alarmins, including IL-1α, and sustains the inflammatory reaction (Figure 1).

Many synthetic adjuvants activate inflammasomes, typically in myeloid cells of the innate immune system. The best characterised inflammasome, NLRP3, is also the most common adjuvant target [56]. NLRP3 expression is induced in myeloid cells via nuclear factor kappa-light-chain-enhancer of activated B-cells (NF-kB) signalling [57], upon recognition of microbial ligands, such as lipopolysaccharide (LPS) or its synthetic mimics. The activation of NLRP3 inflammasome is then triggered upon the loss of cellular homeostasis, for example after membrane damage caused by microbial pore-forming toxins, membrane permeabilising saponins, or during uptake of crystalline compounds such as aluminium crystals or uric acid crystals [58]. NLRP3 can also be activated during local tissue damage upon sensing of DAMPs such as extracellular adenosine triphosphate (ATP), accumulation of reactive oxygen species (ROS) or changes in cell volume [58,59]. NLRP3 does not recognise all of the above signals directly, rather it detects a common downstream consequence of cell damage, such as the efflux of potassium ions, or disruption of the trans Golgi network [60,61]. Unlike NLRP3, other inflammasomes directly recognise specific microbial ligands. For example, NLRC4 detects bacterial flagellin in the cytosol, and it has been assessed as the target of flagellin-adjuvanted vaccines (either admixed or fused to the antigen) [62]. AIM2 detects cytosolic double stranded DNA, and has been identified as a sensor of DNA vaccines [46]. The non-canonical inflammasome, pro-caspase-11, detects cytosolic lipopolysaccharide (LPS) (during infection) or oxidated membrane lipid oxidized 1-palmitoyl-2-arachidonoyl-sn-glycero-3-phosphorylcholine (oxPAPC) (generated during tissue damage) [63]. It also cleaves GSDMD to create the membrane pore and, as a result of potassium efflux, indirectly activates NLRP3 and caspase-1 to cleave IL-1β and IL-18 [63]. Pro-caspase-11 has been assessed as adjuvant target of emulsion-based vaccines [23]. In this review, we will cover adjuvanticity of inflammasome-activating substances with a focus on clinically relevant adjuvants.

## 2. Inflammasome-Mediated Activation of Adaptive Immunity

To understand how inflammasome activating adjuvants can enhance vaccine immunogenicity, it is fundamental to identify the mechanisms of the inflammasome-induced innate signal transduction to adaptive immunity, which eventually leads to pathogen-specific protection. An excellent review by Evavold & Kagan [64] describes how bridging of innate and adaptive immunity via inflammasome signalling is characterised by two elementary features: (1) maturation of innate APCs, which in turn initiate T-cell mediated responses, and (2) secretion of cytokines and chemokines to induce a specific immunological programme [64,65,66]. APC maturation includes antigen loading on MHC molecules, expression of the co-stimulatory molecules CD80 and CD86, and the upregulation and secretion of stimulatory cytokines such as IL-12, which is secreted by DCs to initiate Th1 responses [64,67,68,69,70]. These mechanisms can be seen as a ‘direct bridge’ from innate to adaptive immunity to initiate differentiation of naïve T-cells into effector and memory cells [64]. Additionally, inflammasome-induced activation of adaptive immunity is realised through an ‘indirect route’ via secretion of inflammasome-dependent cytokines, IL-1β and IL-18. Inflammasome-activated pyroptotic cells also release potent DAMPs such as ATP or high mobility group box 1 (HMGB-1) as well as the pro-inflammatory cytokine IL-1α to sustain inflammation. Even though inflammasome-driven cell death eventually terminates cytokine release, recent studies suggest that in some cases, upon inflammasome activation, IL-1β can be released through GSDMD pores from living cells, without pyroptosis, which might represent a novel mechanism for constant and long-lasting secretion of this cytokine [51,71]. 

IL-1α and IL-1β signal via interleukin-1 receptor type 1 (IL-1R1). As IL-1α and IL-1β use the identical receptor, different outcomes depend on the context of their release rather than on ligand-specific effects [72]. IL-1R1 signalling sustains pro-inflammatory innate immune responses by further promoting innate signalling via NF-kB, inducing expression of inflammatory cytokines and chemokines, endothelial adhesion molecules, and leukocyte recruitment [73]. IL-1R1 drives adaptive responses by several mechanisms, including: (I) survival of naïve T-cells through induction of transient IL-2 release, (II) upregulation of the IL-2R, which further enhances survival and proliferation of naïve T-cells, (III) expansion of naïve and memory T-cells and increase of antibody production of B-cells through prolonged T-cell help, (IV) reduction of tolerance through expansion of conventional T-cells in the presence of regulatory T-cells (Tregs), (V) inhibition of cell death in T-cells through the downregulation of Fas ligand, and (VI) differentiation from naïve T-cells into Th17 cells [64,74,75,76,77,78,79,80,81,82]. Even in the absence of CD28 co-stimulation, IL-1β potently drives the differentiation into IL-17/IFN-γ double-producing T-cells (Th17) [83,84] suggesting that IL-1β inducing adjuvants may foster vaccine immunogenicity against pathogens that require Th17 responses, such as pneumococcal or fungal infections. Thus, caspase-1-dependent IL-1 release plays an important role in the transition from innate to adaptive immunity, which is a prerequisite for effective vaccine adjuvants (Figure 2). 

IL-18 signals via the IL-18R, which in combination with IL-12 initiates production of IFN-γ in NK cells, innate lymphoid type-1 cells (ILC1), and Th1 cells, eventually leading to enhanced differentiation into type-I immunity [64,85,86,87,88,89,90,91,92,93]. In the context of adjuvants, IL-18 also mediates antigen-specific CD8 T-cell responses [25,94] (Figure 2). The functions of IL-18 signalling are not always proinflammatory, and depending on the cytokine milieu and genetic background, IL-18 can also promote T helper type 2 (Th2) differentiation in the presence of T-cell receptor activation [95]. 

Caspase-1 cleaves and thereby inactivates IL-33. IL-33 receptor Interleukin-1 receptor-like 1 (ST2) is expressed on innate and adaptive cells, such as alternatively-activated macrophages, type 2 innate lymphoid cells (ILC2), Th2 cells, and Tregs. IL-33 drives Th2 polarisation and tissue repair responses [73], and thus by inactivating IL-33, inflammasomes generally block type-2 responses (Figure 2). IL-33 also augments CD8 T-cell responses by affecting their proliferative and cytotoxic activity [96]. In B-cells, IL-33 mediates B-cell activation of B-1 type cells for enhanced IgM production [97] and also functions as a cell-intrinsic regulator of fitness during the early development of B-cells [98]. 

Not all inflammasome substrates are pro-inflammatory. IL-37 is known to reduce LPS-induced production of IL-1β and tumor necrosis factor alpha (TNF-α) in vitro and thus may have a role in shaping the immune response to adjuvants through inhibition of mitogen-activated protein kinase and NF-kB transcription [99,100]. In support of this, mice expressing human IL-37 transgene developed lower inflammation and were protected from LPS challenge [101]. IL-37 is a less well understood inflammasome substrate, because human IL-37 lacks the mouse orthologue that would enable in-depth genetic and functional studies. 

Together, the caspase-1-dependent cytokines have been shown to influence both the humoral and cellular arms of adaptive responses, and thus may be of relevance for immunogenicity provided by inflammasome-activating adjuvants. Inflammasomes may be of particular interest for vaccines targeting cell-mediated immunity [56], as this arm of immune response is the direct place of action for inflammasome-dependent cytokines.

## 3. Clinically Relevant NLRP3-Activating Adjuvants

### 3.1. Aluminium Salt-Based Adjuvants

Aluminium (Al) salts, usually referred to as ‘alum’, were the first reported clinical adjuvants and were widely used long before the inflammasome was discovered by Jürg Tschopp and colleagues [54]. Chemically, alum is a hydrated sulphate salt of aluminium, originally employed in antigen (Ag) precipitation and not used as an adjuvant today. However, in the context of vaccine adjuvants, the typically used salts Al-hydroxide and Al-phosphate are commonly referred to as ‘alum’, often without making a distinction on the type of salt used. This, in part, is the reason for the ongoing ambiguity in understanding the MoA of Al-based adjuvants [103]. Al-salts are usually associated with the antigen through electrostatic attraction, hydrophobic attraction or ligand exchange [104]. 

In terms of MoA, one of the most studied Al-based adjuvants is Imject, a commercial adjuvant developed by Thermo Scientific, a formulation comprising Al-hydroxide and Mg-hydroxide. In a study in 2008, NLRP3 was identified as a sensor for Imject crystal-induced lysosomal damage, in vitro, albeit requiring LPS-induced priming via TLR4 for robust IL-1β secretion [7]. 

Three independent studies showed strongly diminished immunogenicity in *NLRP3^−/−^* mice after immunisation with Imject-adjuvanted Ovalbumin (OVA) [6,8,9]. Two studies suggested that macrophages and mast cells sense both Imject and Alhydrogel (Al-hydroxide, developed by Brenntag, Germany) [13], inducing the release of host DNA serving as DAMP that mediates the adjuvant activity [12]. However, a more recent study highlighted differences in the immunostimulatory properties between Imject, Alhydrogel, and an alum-precipitated Ag, bringing into question the extrapolation of Imject data to the clinically used Al-adjuvants [105]. A study using human serum albumin (HSA) as the antigen found that the NLRP3 inflammasome is dispensable for the adjuvant activity of Al-hydroxide [11]. Later studies confirmed that Al-hydroxide adjuvanticity is independent of NLRP3 and caspase-1 in OVA immunisation models [12,13,14], with further in vitro studies reporting that crystalline structures such as Al-hydroxide or uric acid cross-link DC membrane lipids, activating the spleen tyrosine (Syk) kinase, and mediating adjuvanticity in an inflammasome-independent fashion [106,107]. 

Differential results may result from divergent environmental conditions such as the presence of diverse microbiome patterns, which are known to interact with inflammasomes [108]. In addition, the use of different Al-hydroxide compositions in varying concentrations may result in altered adaptive immune responses. Although a divide remains on the ability of Al-hydroxide to activate the NLRP3 inflammasome in vivo, there is growing evidence that the NLRP3 inflammasome is dispensable for high-level antibody responses induced by Al-hydroxide [15]. In humans, IL-1β blockade using canakinumab did not reduce antibody responses to Al-hydroxide-adjuvanted meningococcal vaccine (Menjugate™, GSK, Sienna, Italy) in healthy volunteers [109]. Thus, Al-hydroxide has a questionable benefit in meningococcal conjugate vaccines and today the licensed quadrivalent meningococcal conjugate vaccines do not contain Al adjuvants [110].

In addition to the ‘classic’ Al-salt based adjuvants, structurally-modified Al particles are being assessed for their ability to activate NLRP3. Recently, Sun et al. generated ‘aluminium oxyhydroxide nanorods’ with a defined surface functionalisation and charge, which enhanced NLRP3 activation in vitro and increased antibody production in an OVA immunisation model, compared to Al-hydroxide microparticles [111]. Al (oxy)hydroxide nanoparticles increase the production of uric acid and thereby enhance NLRP3 activation compared to microparticles, further indicating a relevance of physical characteristics for the adjuvanticity of Al particles [112]. In line with this, Orr et al. demonstrated that, compared to the conventional Al-hydroxide adjuvant (Alhydrogel), consisting of aggregates of particles in various sizes, newly developed stable nanoparticles (“nanoalum”) provide improved immunogenicity against lethal influenza challenge in mice [113], dependent on NLRP3, ASC, and IL-18R.

Of note, an earlier study already demonstrated that only positively-charged particulate vaccine adjuvants such as Al-hydroxide (Alu-Gel-S) or polymer-chitosan nanoparticles (CNPs) activate the NLRP3 inflammasome [10]. Together, these results indicate that the full potential of Al-salt-based adjuvants is not yet reached, as physical modification of the size or shape might improve adjuvanticity, possibly through enhanced activation of the NLRP3 inflammasome.

Al hydroxide is also used in combinatorial adjuvant systems such as AS04 in order to target multiple pathways [103,114,115]. In the licenced vaccines against the human papillomavirus (HPV, Cervarix™ GSK, Rixensart, Belgium) and hepatitis B virus (HBV, Fendrix™, GSK, Rixensart, Belgium), AS04 consists of 3-O-desacyl-4′-monophosphoryl lipid A (MPL-A), an LPS-derivative that activates TLR4, adsorbed onto Al-hydroxide [115]. Inclusion of MPL-A induces robust TLR4 mediated NFκB activation, resulting in APC activation, production of the NFκB targets TNF-α and IL-6, and Ag-specific T-cell activation [115]. Targeting TLR4 via MPL-A improved clinical applications, as AS04-adjuvanted HPV and HBV vaccines induced higher antibody responses compared to vaccines adjuvanted with Al-hydroxide alone [22,103,116]. Since TLR priming represents a prerequisite for robust inflammasome activation in vitro, it may seem plausible that TLR4 priming by MPL-A also enhances inflammasome activation in vivo. However, whether AS04 provides improved vaccine responses through increased NLRP3 activation remains to be assessed.

### 3.2. Saponin-Based Adjuvants

Saponins have been recognised as vaccine-adjuvanting substances for several decades [117]. Their adjuvanticity is based on disrupting the membrane integrity and the induction of danger signals [114,118]. The most widely used substance from the saponin family is QS-21, a specific triterpene saponin fraction purified from *Quillaja saponaria*. Adjuvant systems AS01 and AS02, developed by GSK, contain QS-21 in combination with the TLR4 agonist MPL-A and liposomes, or MPL-A in an oil-in-water emulsion, respectively [119]. AS01 especially is of clinical relevance, as part of the malaria vaccine Mosquirix™ (currently in pilot introduction in Africa, GSK, Rixensart, Belgium) and the licenced vaccine against herpes zoster, Shingrix™ (GSK, Rixensart, Belgium). Recently, Marty-Roix et al. clearly demonstrated that clinical grade QS-21 activates the NLRP3 inflammasome in an ASC-, TLR4-, myeloid differentiation primary response 88 (MyD88)-, and TIR-domain-containing adapter-inducing interferon-β (TRIF)-dependent manner in murine bone marrow-derived DCs (BMDCs) and suggested phagocytosis, followed by lysosomal acidification, as a possible mechanism [20]. Furthermore, they showed that caspase 1/11 double-deficient BMDCs and macrophages exhibit strongly reduced IL-1β secretion upon stimulation with QS-21. These results raise the question whether QS-21 targets the NLRP3/caspase-1 axis directly or via the non-canonical inflammasome (caspase 11 in mice, caspase 4/5 in humans), which induces NLRP3-dependent caspase-1 activation by promoting potassium efflux [120,121,122]. Importantly, QS-21-mediated NLRP3 activation requires co-stimulation with MPL-A, indicating that TLR priming is a prerequisite for robust inflammasome activation by QS21 in vitro [20]. In contrast, in vivo immunization with QS-21-adjuvanted HIV-1 gp120 showed that QS-21 indeed enhances antibody and T-cell responses in mice but NLRP3 deficiency boosted these effects, indicating that NLRP3 might decrease QS-21-induced Ag-specific responses in vivo [20]. In human monocyte-derived DCs, cholesterol-dependent endocytosis of QS-21 formulated in liposomes induced lysosomal membrane permeabilisation and activated DCs in a Syk kinase- and cathepsin B-dependent manner [21]. This lysosomal destabilization may in parallel induce activation of the NLRP3 inflammasome. A further study demonstrated that immunisation with QS-21 in liposomes leads to an enrichment of QS-21 in CD169^+^ resident macrophages in the dLN, which activates caspase-1 and results in innate cell recruitment, DC activation, and T-cell priming [19]. In contrast to the results of Marty-Roix et al. [20], T-cell responses were diminished and antibody responses were unaltered in caspase-1 deficient mice after immunisation with QS-21/liposome formulation [19]. The caspase-1-dependent adjuvanticity of QS-21 in liposomes was partially mediated through the release of HMGB1 and reliant on MyD88, indicating that TLR priming is required for robust QS-21-mediated immunogenicity in vivo. This study also used caspase-1/11 double-deficient mice, which do not allow to distinguish between canonical and non-canonical NLRP3 activation. However, membrane permeabilization and DAMP release as proposed mechanisms indicate canonical NLRP3 activation. Taken together, QS-21 is a potent activator of the NLRP3 inflammasome, although it requires additional immunostimulatory substances to exhibit its full adjuvanting potential. Hence, adjuvant systems such as AS01, in which QS-21 and MPL-A synergistically activate de novo pathways and induce robust antibody and T-cell responses, represent a sophisticated approach to improve the immunogenicity of modern vaccines [123,124,125]. 

Quil A, which is a mixture of saponins extracted from *Quillaja saponaria*, is a more heterogenous saponin adjuvant and is part of the immunostimulating complex-based adjuvants ISCOM and ISCOMATRIX (IMX) [24,126]. Besides Quil A, these formulations contain phospholipids and cholesterol, forming spherical, cage-like structures 40 nm in diameter. Vaccines formulated with these adjuvants provide long-lasting antibody responses, a balanced Th1/Th2 response, and generation of cytotoxic T-cells [127,128,129]. In 2011, Duewell et al. demonstrated that OVA, adjuvanted with IMX, induces high levels of IL-1β, IL-6, granulocyte-macrophage colony-stimulating factor (GM-CSF), and IL-12p40 as well as low levels of IL-4 and IL-5 in the dLNs, which indicates early inflammasome activation and a mixed Th1/Th2 type response [24]. Furthermore, the authors verified caspase-1 dependency of OVA/ISCOM-induced IL-1β secretion in vitro. However, a later study assessed inflammasome activating characteristics of IMX and highlighted substantial differences between in vitro and in vivo mechanisms of vaccine adjuvants [25]. Wilson et al. observed IMX-induced IL-1β production in dLN in vivo and robust NLRP3-, ASC-, and caspase-1/11-dependent IL-1β secretion in LPS-primed thioglycollate-induced peritoneal macrophages in vitro, suggesting dependence on lysosomal destabilisation [25]. In contrast, the innate NK cell response to IMX was dependent on IL-18R but independent of NLRP3 and IL-1R1 in vivo. In line, the IL-18 pathway was crucial for IMX-OVA-induced adaptive immunity, as *IL-18^−/−^* and *IL-18R^−/−^* mice showed strongly diminished levels of Ag-specific CD8 T-cells and IgG2c antibodies, while deficiency in IL-1R1 or NLRP3 had no effect on adaptive immune responses. Interestingly, TNF-α provided a physiological inflammasome priming signal, which might substitute for the lack of a TLR agonist in IMX. Together, this study demonstrated inflammasome-dependent and -independent IMX-induced mechanisms and raised the question of how IL-18 contributes to IMX-induced immunity in an NLRP3-independent fashion. Potentially, constitutively expressed IL-18 could be released via APC cell death at the site of injection, or alternatively other proteases such as caspase-8, able to process IL-18 [130], could substitute for inflammasome/caspase-1-mediated maturation of pro-IL-18 [25]. 

Recently, Cibulski et al. assessed adjuvant characteristics of alternative saponin formulations and replaced Quil A in ISCOM and IMX by fractions extracted from *Quillaja brasiliensis* to generate *Quillaja brasiliensis* fraction 90 (QB-90) and ISCOMATRIX-like *Quillaja brasiliensis* fraction 90 (IMXQB-90), respectively [128,131]. Inoculation of mice with QB-90 or IMXQB-90 induced antibody production, recruitment and activation of myeloid and lymphoid cells in spleen and dLN, and production of the Th1-type cytokines IFN-γ and TNF-α. In BMDCs, both saponin formulations induced caspase-1-dependent IL-1β production in vitro, which is in line with the above-mentioned studies of IMX. However, whether the inflammasome mediates the potential adjuvanticity of QB-90 or IMXQB-90 in vivo, remains elusive. 

In summary, the literature clearly demonstrates that saponin-based adjuvants activate the NLRP3 inflammasome pathway to initiate an early IL-1β- or IL-18-mediated innate immune response, but the impact of this mechanism on adjuvanticity remains a matter of debate. Of note, saponins are usually used in combination with other components such as TLR agonist or liposomes, and the observed innate activation is likely a result of additive and/or synergistic effects of the individual components.

### 3.3. Emulsion-Based Adjuvants

In the context of adjuvants, emulsions commonly include water-in-oil (W/O) and oil-in-water (O/W) emulsions such as the prototypes Montanide 720 and MF59, respectively [4,132,133]. Double emulsions such as water-in-oil-in-water (W/O/W) and oil-in-water-in-oil (O/W/O), able to induce biphasic responses, have also been developed for veterinary applications [134,135]. Modern clinically relevant O/W emulsions, such as MF59 and AS03 - which additionally contains α-tocopherol (vitamin E)-use the fully metabolisable lipid squalene, produced during cholesterol synthesis in humans [103,136]. They have been used in vaccines against pandemic influenza H1N1 and avian influenza H5N1 [103,119] and in vaccine development for age-specific groups such as the children and elderly [137,138]. MF59 and AS03 have been shown to recruit CD11b^+^/MHC-II^+^ cells and induce IL-12(p40)/IL-5 expression (MF59) or to recruit monocytes/granulocytes and induce C-C motif chemokine 2 (CCL2), CCL3, IL-6, CSF3, and C-X-C motif chemokine ligand 1 (CXCL1) (AS03) [16,114,139,140]. Among other cytokines and pro-inflammatory mediators, expression of the inflammasome related genes *caspase-1*, *IL-1β*, and *IL-1R1* are strongly induced by MF59 [16]. Furthermore, MF59 adjuvanticity depends on the TLR adaptor molecule MyD88 [15], and induces the release of ATP, which can activate the NLRP3 inflammasome via the P2X7 receptor [17]. Nevertheless, two independent studies demonstrated that NLRP3 is not required for the adjuvanticity of MF59, despite of the necessity for ASC, which also functions as inflammasome adaptor molecule [15,18]. There are no publications addressing NLRP3 inflammasome activation by the adjuvant system AS03; this might be due to the lack of appropriate TLR-mediated priming by these emulsions, since GLA-SE, a combination of the synthetic TLR4 agonist GLA and squalene oil-in-water emulsion, mediates adjuvanticity through NLRP3 activation and TLR4 signalling, inducing robust Th1 and B-cell responses [23]. In contrast, squalene oil-in-water emulsion alone induced considerably lower adjuvanticity when compared to GLA-SE [23,94]. Interestingly, besides caspase-1, GLA-SE-mediated adjuvanticity is also dependent on caspase-11, the non-canonical inflammasome [23]. Altogether, the studies above indicate that squalene oil-in-water emulsion might be an inducer of canonical and non-canonical NLRP3 inflammasome activation, if supplemented with a TLR agonist.

### 3.4. TLR Agonists as Adjuvants

TLRs represent a prominent vaccine adjuvant target due to their potent activation of MyD88 and TRIF-mediated activation of NFκB and interferon-regulatory factors. Thereby, TLR-agonists can provide the “first step” priming signal for robust NLRP3 inflammasome activation. LPS-analogues, and thus TLR4 agonists, such as MPL-A or GLA, have been used in the above mentioned combinatorial vaccine adjuvants AS01, AS02, AS04, and GLA-SE to enhance adjuvanticity of QS-21, Al particles, and squalene O/W emulsion. Of note, in addition to TLR4, the non-canonical inflammasome represents an intracellular sensor of LPS in vivo [141], which has not yet been confirmed for MPL-A. MPL-A alone does not induce caspase-1 activation and IL-1β secretion in vitro [20,21,142]. This is in line with LPS, which only provides transcriptional upregulation of pro-IL-1β and NLRP3 through priming and requires a second inflammasome stimulus for robust caspase-1 activation, in vitro [7,57,143]. In contrast, LPS alone is sufficient for IL-β secretion in vivo [141,144], which might also indicate that MPL-A is a potential inducer of IL-1β secretion in vivo. Interestingly, immunisation with the synthetic TLR4 agonist GLA induced the production of small amounts of IL-1β in vivo [23]. However, there are mechanistic differences between LPS and MPL-A, since LPS induced signalling is predominantly transmitted via MyD88, whereas MPL-A is a TRIF-biased agonist of TLR4, which might explain its reduced toxicity [145]. Taken together, MPL-A has an immunostimulatory effect and enhances adjuvanticity of combinatorial adjuvants, but it remains unclear whether it triggers the same pathway as LPS in vivo.

Among others, adjuvants targeting TLR3 (dsRNA analogues such as poly(I:C)), TLR5 (flagellin), TLR7/TLR8 (imidazoquinolines such as Imiquimod), or TLR9 (CpG oligodeoxynucleotide) have been assessed for adjuvanticity [3]. Polyinosinic:polycytidylic acid (Poly-IC) can generate a comprehensive immune response, which is suited for anti-viral and tumour vaccines as it activates APCs and induces memory T and B-cell production [146]. Flagellin may also provide excellent adjuvanticity as it induces cellular and humoral immune responses through activation of two signalling pathways via TLR5 and NLRC4 [62]. Similarly, TLR7/8 agonists, such as Imiquimod have been shown to activate APCs, induce humoral and cellular immunity, and especially enhance Th1 responses [147]. Despite safety concerns regarding the production of autoreactive antibodies, TLR9 agonists have been shown to induce robust immune responses to the vaccine antigen with no apparent adverse reactions [3,148], and have been clinically evaluated with a number of infectious disease and cancer vaccines [149]. This includes CpG-adjuvanted vaccines against anthrax, hepatitis B, malaria, and influenza [149]. Clinical trials using CpG in vaccines against anthrax [150,151] and HBV [152] have been especially promising as they demonstrated rapid and persistent production of protective antibodies, even in immunocompromised HIV-infected individuals [153,154]. Recently, Kim et al. showed that in vitro, CpG induces the expression of NLRP3, ASC, caspase-1, and IL-1β in a TLR9-dependent fashion, suggesting that the NLRP3 inflammasome could be involved in the mechanism of CpG-adjuvanted vaccines [26].

In summary, TLR agonists can enhance immunogenicity through the activation of different PRRs, eventually initiating activation of NFκB signalling, which induces a pro-inflammatory signalling cascade and may provide a priming signal for NLRP3 inflammasome activation in combinatorial adjuvants. However, in case of the TLR5 agonist flagellin, NLRC4 inflammasome activation is provided by the same single adjuvant.

## 4. Experimental Adjuvants That Activate Inflammasomes

In addition to the above inflammasome-activating adjuvants, of which the majority are commonly used in clinical applications, other substances have been assessed for inflammasome-mediated adjuvanticity. These include compounds that have been shown to activate NLRP3, NLRC4, AIM2, pyrin, and the non-canonical inflammasome. 

### 4.1. NLRP3 Inflammasome

Chitosan is a biocompatible and biodegradable cationic polymer obtained from chitin [56]. It has a good safety profile and has been shown to promote adjuvanticity through induction of Ag-specific IgG1/IgG2a and Th1/Th2/Th17 responses [155]. Chitosan activates NLRP3-dependent IL-1β secretion in BMDCs, bone marrow-derived macrophages (BMDMs), and human peripheral blood mononuclear cells [33,34,36]. In addition to the NLRP3 inflammasome pathway, chitosan also activates the cyclic GMP-AMP synthase (cGAS) - stimulator of interferon genes (STING) pathway to promote cellular immunity [34]. In combination with CpG, chitosan-promoted Ag-specific Th1, Th17, and IgG2 responses were strongly dependent on the NLRP3 inflammasome in vivo [36]. Adjuvant characteristics of chitosan have also been assessed in combination with Al-salts. Chitosan-aluminium sulphate nano-particles, as well as conventional chitosan particles, induced IL-1β production in BMDCs in an NLRP3- and ASC- dependent manner [35]. 

The synthetic adjuvant trehalose-6,6′-dibehenate (TDB, an analogue of mycobacterial cord factor trehalose-6,6′-dimycolate (TDM)) [156], promotes Syk and CARD9-dependent activation of innate immunity through the C-type lectin Mincle, which mediates adjuvanticity via induction of robust Th1 and Th17 responses [157,158,159]. Importantly, TDB initiates NLRP3-dependent caspase-1 processing and IL-1β secretion in BMDCs, which relied on potassium efflux, lysosomal rupture, and ROS production [30]. Furthermore, TDB-induced recruitment of neutrophils was strongly impaired in *NLRP3^−/−^* mice [30]. A further in vivo study, which used TDB formulated into liposomes, demonstrated that TDB induces MyD88- and IL-1R1-dependent Th1/Th17 responses independent of IL-18 and IL-33 signalling [29], further supporting the NLRP3 inflammasome as an essential mediator that contributes to TDB-induced adjuvanticity. 

Calcium phosphate nanoparticles (CaP-NPs) have been demonstrated as a promising adjuvant candidate due to their ability to induce balanced Th1 and Th2 immune responses as well as their high degree of biocompatibility and biodegradability [160]. Initially, He et al. showed in a herpes simplex virus type 2 (HSV-2) challenge model that CaP-NPs induce higher titres of IgG, lower titres of IgE, and improved protection against HSV-2 compared to Al-hydroxide [161,162]. A later study demonstrated that CaP crystals induce NLRP3-, ASC-, and caspase-1-dependent secretion of IL-1β in vitro [28]. However, antibody responses after immunisation with another form of calcium phosphate - hydroxyapatite - particles were shown to be independent of NLRP3, ASC and caspase-1, indicating that CaP particles might provide NLRP3 inflammasome-independent adjuvanticity in vivo [27].

Biodegradable and non-degradable nano- and microparticles, have been used for Ag delivery and as immunopotentiators [56,163]. Biodegradable poly(lactic-co-glycolic acid) (PLGA) microparticles have been studied as vehicles for antigens (e.g., Hepatitis B surface antigen (HBsAg), tetanus toxoid, HIV gp120) and TLR-agonist adjuvants, and have been reported to induce humoral immunity through the induction of Ag-specific IgG1 and IgE [56,164,165]. Sharp et al. showed that PLGA induces NLRP3- and caspase-1-dependent IL-1β release in vitro [31]. In vivo, PLGA induced IL-6 secretion and innate immune cell recruitment in an NLRP3-dependent manner, but the production of Ag-specific antibodies was independent of the NLRP3 inflammasome [31], putting the relevance of NLPR3 in PLGA-mediated adjuvanticity into question. 

Biocompatible silica particles induced NLRP3-dependent IL-1β and HMGB release in human lung epithelial cells [38]. Furthermore, depending on the size, silica particles induced caspase-1-dependent IL-1β secretion in BMDMs, and IL-1β-mediated lung inflammation in mice [37]. Interestingly, Kuroda et al. suggested that silica-induced lysosomal damage also activates the NLRP3-independent PGE2-inducing pathway, at least in addition to the NLRP3 inflammasome pathway [166]. 

Moreover, gold nanoparticles, which have been used in cancer vaccines [56,167,168], might also activate the inflammasome, as particles with a distinct size and shape have been shown to induce IL-1β and IL-18 secretion in BMDCs [39].

Taken together, the general role of NLRP3 in particle-induced (including particulate Al-adjuvants) immune responses, and especially regarding adjuvanticity in vivo, remains elusive.

Vaults, which are large cytoplasmic ribonucleoprotein particles that contain three proteins and a small untranslated RNA [32,169,170], have been shown to display self-adjuvanting properties [171,172,173]. Zhu et al. engineered vaults that contain the immunogenic *Chlamydia trachomatis* epitope PmpG-1 and showed that PmpG-1-vaults induce NLRP3-, ASC-, and caspase-1-dependent IL-1β secretion in human monocytic (THP-1) cells, and PmpG-1 responsive CD4^+^ T-cells after immunisation in mice [32]. However, whether adjuvanting vaults promote NLRP3-mediated adjuvanticity in vivo, remains unclear. 

### 4.2. NLRC4 Inflammasome

The NLRC4 inflammasome is, in addition to the transmembrane receptor TLR5, an intracellular sensor of bacterial flagellin [55]. Flagellin has been positively evaluated as a broad-spectrum vaccine adjuvant capable of inducing potent systemic and mucosal adaptive immune responses [62,174]. In humans it has been shown to provide robust antigen-specific humoral immunity to influenza vaccines and is well tolerated [175,176]. In contrast to NLRP3, which is activated through numerous different stimuli that induce cellular stress, NLRC4 specifically senses flagellin via NLR family, apoptosis inhibitory protein 5 (NAIP5) and NAIP6. This raises the question of whether NLRC4 or TLR5 (or both) are required for robust flagellin-induced adjuvanticity rather than flagellin-mediated NLRC4 activation in general. Using *Naip5^−/−^* and *TLR5^−/−^* mice on a C57/BL6 background, López-Yglesias et al. demonstrated that IgG2c responses against flagellin are TLR5- and NAIP5-dependent, whereas the dominant IgG1 responses were only partially dependent. Flagellin-specific IgG1 response was also mediated by a TLR5-, NAIP5-, and MyD88-independent pathway. Interestingly, flagellin induced a codominant IgG1 and IgG2a response in A/J mice. After immunisation with flagellin-adjuvanted OVA, mice only developed robust OVA-specific IgG1 responses, which were MyD88-dependent and independent of TLR5, NAIP5, and caspase-1. However, deficiency in both TLR5 and caspase-1 led to reduced IgG1 responses [41]. These results indicate that flagellin-induced adjuvanticity is mediated through a third TLR5- and NLRC4-independent pathway but requires at least one of the specific sensors TLR5 or NLRC4 to display its full potential. This is in line with another study that showed equal OVA- and flagellin-specific IgG1 responses between WT, *TLR5^−/−^*, and *NLRC4^−/−^* mice, but strongly diminished antigen specific responses in TLR5/NLRC4 double-deficient mice [44]. Interestingly, flagellin could be a promising alternative to NLRP3-activating adjuvants in vaccines for immunocompromised patients, as it properly induces inflammasome activation in DCs from HIV patients that harbour intrinsic NLRP3 defects [177]. Flagellin has also been assessed for adjuvanticity after administration of a recombinant modified vaccinia virus Ankara (rMVA) vaccine encoding the flagellin gene. Immunisation with rMVA-flagellin induced enhanced secretion of mucosal IL-1β and TNF-α, resulting in elevated T-cell and antibody responses, which were diminished in *NLRC4^−/−^* mice [43]. Similarly, incorporating flagellin into the replicon of an alphavirus enhanced IgG1 and IgG2a/c titres, indicating an enhancement of Th1 and Th2 type responses. The adjuvanticity of the flagellin-expressing alphavirus was partially dependent on TLR5 [40]. In addition to virus-based delivery, flagellin has also been administered as a DNA-plasmid encoding the flagellin gene. After co-immunisation with a second OVA-encoding plasmid, mice showed Ag-specific antibody responses and MHC Class I-dependent cellular immune responses [42]. Of note, adjuvant delivery via DNA molecules might also activate the AIM2 inflammasome.

### 4.3. AIM2 Inflammasome

The AIM2 inflammasome is a sensor that detects intracellular DNA, which can originate from intracellular pathogens (PAMPs), or endogenous affected cells (DAMPs) that have lost nuclear envelope integrity [178]. Thus, it is not very surprising that AIM2 has been identified as a sensor of DNA vaccines, in which the Ag is produced in vivo by the endogenous transfected cells [46]. Here, the DNA plasmid itself represents an intrinsic adjuvant, which enhances immunogenicity towards the vaccine-encoded immunogens [46]. However, DNA vaccines have also been used in combination with established adjuvants such as Al-salts or CpG oligonucleotides [179]. Today, there are no DNA vaccines approved for use in humans but clinical trials using DNA vaccines against HIV and Hepatitis B have demonstrated potential immunogenicity [179]. DNA vaccine-induced humoral and cellular Ag-specific adaptive responses rely on the AIM2 inflammasome, but surprisingly not on IL-1R1 and IL-18R [46]. Interestingly, IFN-α/β were reduced in Aim2-deficient mice after DNA vaccination, which might indicate that DNA vaccines induce IFN-α/β signalling, thereby promoting dispensability of IL-1/IL-18 signalling [46]. Recently, two studies demonstrated that DNA vaccines encoding the *Aim2* gene itself might be a promising approach to enhance immunogenicity [45,47]. 

### 4.4. Pyrin Inflammasome

The pyrin inflammasome senses inactivating modifications of the Rho GTPase (RhoA), which regulates cytoskeletal remodelling and is a frequently used pathogen entry route. Therefore, pyrin can indirectly sense so-called “homeostasis altering molecular processes” (HAMPs), which does not rely on the detection of conserved molecules and thus facilitates sensing of evolutionary novel infections [180,181]. Cholera toxin B (CTB) from *Vibrio cholerae* can be administered as a vaccine adjuvant in various forms such as Ag-fusion-protein, co-administered with an Ag, chemically coupled to an Ag, or as a DNA vaccine [182]. Cholera toxin has been shown to induce caspase-1 and IL-1R1-dependent Th17 responses in human monocytes, which upregulated expression of the inflammasome-related genes *Nlrp1*, *Nlrp3*, and *Nlrc4* [48]. However, another study showed that NLRP3 activation differs between different strain biotypes of toxins secreted from *Vibrio cholerae* [50]. Recently, Orimo et al. showed that CTB induces IL-1β production in peritoneal macrophages through the NLRP3 and pyrin inflammasome [49]. Here, protein kinase A has been suggested to mediate RhoA phosphorylation in CTB-induced macrophages, which results in pyrin inflammasome-mediated IL-1β production [49]. Thus, the pyrin inflammasome may contribute to adjuvanticity of CTB. 

### 4.5. Non-Canonical Inflammasome

The non-canonical inflammasome is represented by murine caspase-11 and human caspase-4/5, which sense intracellular LPS [141,183]. Importantly, Zanoni et al. identified oxPAPC as a further ligand of caspase-11 [52]. oxPAPC “hyperactivates” living DCs via caspase-11, independent of TLR4 in an CD14-dependent manner, which results in non-canonical NLRP3 inflammasome activation [52,53]. In contrast to LPS, oxPAPC binds caspase-11 via a different domain, which induces IL-1β/IL-18 release but not pyroptosis. Of note, GSDMD is the regulator that mediates oxPAPC-induced IL-1 secretion from living cells [51]. It is currently unknown why GSDMD does not induce pyroptosis downstream of oxPAPC. The mechanism may involve active repair of GSDMD pores in living cells, but this remains to be tested [184]. This mechanism of hyperactivated APCs, which induces strong innate immune responses in absence of pyroptotic cell death might also activate adaptive immunity through long-lasting pro-inflammatory signalling. Indeed, using an OVA immunisation model, Zanoni et al. demonstrated that oxPAPC potentiates adaptive immune responses in a caspase-11-dependent manner [52]. Thus, oxPAPC is a promising candidate for future vaccine adjuvants, which may provide superior immunogenicity over conventional adjuvants such as Al salts, through hyperactivation and the lack of pyroptosis.

## 5. Conclusions

This review of the available literature highlights the complexities of evaluating and ascertaining the MoA of vaccine adjuvants. Adjuvants include a broad spectrum of compounds, often used in non-identical formulations between different laboratories, adding to the diversity of observations in their MoA. Frequently, in vivo experiments revealed substantial discrepancies in the observed MoA compared to previous in vitro analyses. Although in vitro studies are well suited to identifying adjuvant targets and generating hypotheses, we conclude that robust in vivo studies using clearly defined adjuvant formulations are indispensable for generating reliable data of adjuvant mechanisms of action. For in vivo studies, it is also important to include readouts of cell-based immunity, in addition to measurements of antibody production, as T-cells are the direct place of action for inflammasome-dependent cytokines. It must be noted however, that, although murine models represent an essential element in the pre-clinical development of novel vaccines, translatability to the clinic is often limited, due to the many differences on the physiological, cellular and molecular level between the species [185]. In this context it is of paramount importance to thoroughly evaluate safety profiles of new inflammasome activating adjuvants, since inappropriately strong inflammasome activation may induce IL-1β mediated inflammatory symptoms, as seen in auto-inflammatory diseases such as Familial Mediterranean fever or cryopyrin-associated periodic syndromes [186].

Depending on the specific adjuvant, inflammasomes can contribute to immunogenicity as an important bridge between innate and adaptive immunity. In particular, the NLRP3 inflammasome is involved in the MoA of established vaccine adjuvants, although the precise impact of NLRP3 activation on their adjuvanticity requires further evaluation. The combination of NLRP3-activating substances with various TLR agonists, which provide a robust priming signal through upregulation of pro-inflammatory genes, facilitate greatly enhanced innate immune responses and higher adaptive responses, resulting in robust and persistent protection from the addressed pathogen. Therefore, combinatorial NLPR3-activating vaccine adjuvants are a promising approach in advancing clinical vaccines. Other inflammasome such as NLRC4 also represent a promising target of vaccine adjuvants as they also induce caspase-1-mediated secretion of IL-1-family cytokines but require a more specific and tangible trigger with a potentially clearer MoA.

A drawback of inflammasome-activating adjuvants might be the induction of caspase-1-mediated pyroptosis, which although pro-inflammatory may also terminate immunostimulation as dying immune cells cannot provide persistent inflammatory signalling. Here, oxPAPC, an activator of the non-canonical inflammasome, may be capable of circumventing this issue as it induces the release of pro-inflammatory cytokines in absence of pyroptosis. Thus, oxPAPC-induced hyperactivation of APCs could be a promising concept for future vaccines.

In summary, inflammasomes have emerged as highly relevant mediators of the MoA of a number of vaccine adjuvants, and their engagement should be actively considered during “intelligent” vaccine design and the development and evaluation of novel immunostimulatory compounds for combating current and emerging pathogens. 

## Figures and Tables

**Figure 1 vaccines-08-00554-f001:**
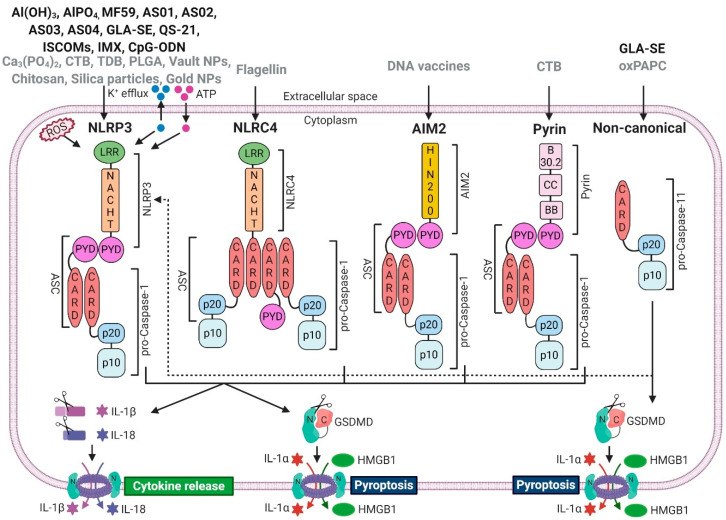
Vaccine adjuvants can activate inflammasomes. Clinically approved (black) and experimental (grey) vaccine adjuvants induce (or possibly induce) assembly and activation of canonical- (NLRP3, NLRC4, AIM2, Pyrin) and non-canonical (caspase-11 in mice, caspase-4/5 in humans) inflammasomes. Canonical inflammasomes recruit caspase-1 via the adaptor molecule ASC, which leads to proximity-induced auto-processing and activation of caspase-1, resulting in cleavage of pro-IL-1β and pro-IL-18 as well as the secretion of their mature forms. Activated caspase-1 also cleaves and activates GSDMD, resulting in pore formation and pyroptosis, which mediates the release of pro-inflammatory DAMPs such as IL-1α or HMGB1. Caspase-11 cleaves GSDMD and induces pyroptosis, but it does not process pro-IL-1β or pro-IL-18. However, non-canonical inflammasomes activate the NLRP3 inflammasome, which indirectly induces the maturation and secretion of IL-1β and IL-18 via the non-canonical route. Abbreviations: apoptosis-associated speck-like protein containing a CARD (ASC), caspase recruitment domain (CARD), pyrin domain (PYD), high-mobility group box 1 (HMGB1), leucine-rich repeat (LRR), domain present in NAIP, CIITA, HET-E, and TP1 (NACHT), gasdermin D (GSDMD), reactive oxygen species (ROS), adenosine triphosphate (ATP), coiled-coil (CC), exon B30.2 domain (B30.2), B-box-type zinc finger domain (BB); immunostimulating complex (ISCOM); ISCOMATRIX (IMX). Created with BioRender.com (Toronto, Canada).

**Figure 2 vaccines-08-00554-f002:**
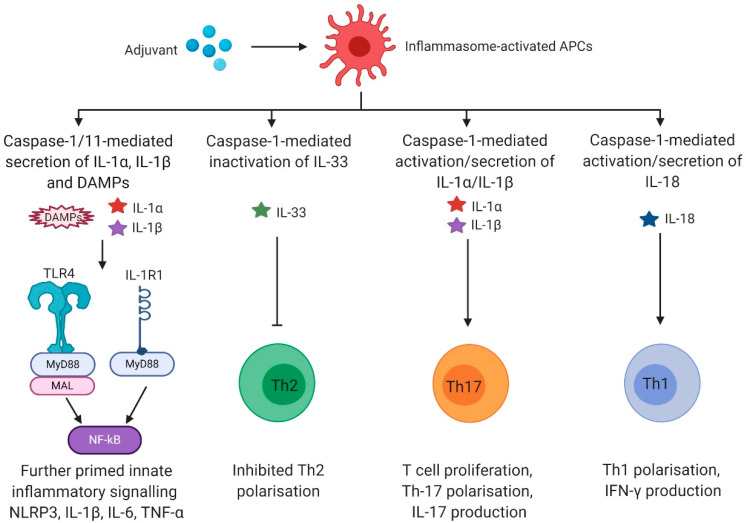
Caspase-1-dependent cytokines activate adaptive immunity. IL-1α and IL-1β signalling via IL-1R1 activates the pro-inflammatory transcription factor NFκB via the MyD88 pathway. In addition, caspase-1 induces NFκB activation through the cleavage of MAL [102]. NFκB activation results in the production of various pro-inflammatory cytokines such as IL-6 or TNF-α, which (I) enhances innate and adaptive immune responses and (II) provides a further priming signal for robust inflammasome-mediated signalling. Cleavage and inactivation of the potent Th2 driver IL-33 by active caspase-1 blocks Th2 polarisation. Mature IL-1α and IL-1β directly affect lymphoid cells by promoting differentiation from naïve T-cells into Th17 cells, T-cell survival, or increased antibody production of B-cells through prolonged T-cell help. After caspase-1-mediated activation, IL-18 binds IL-18R and thereby initiates the production of IFN-γ eventually inducing enhanced differentiation into Th1 type immune cells. Abbreviations: antigen presenting cells (APCs), danger-associated molecular pattern (DAMP), Toll-like receptor (TLR), interleukin 1 receptor 1 (IL-1R1), myeloid differentiation primary response 88 (MyD88), Myd88 adapter-like (MAL), nuclear factor kappa-light-chain-enhancer of activated B-cells (NFκB), nucleotide-binding oligomerisation domain, leucine rich repeat and pyrin domain containing protein 3 (NLRP3). Created with BioRender.com (Toronto, Canada).

**Table 1 vaccines-08-00554-t001:** Overview of inflammasome-activating adjuvants.

Adjuvant	Description	Targeted Inflammasome	References
Al(OH)_3_, AlPO_4_	Al-hydroxide, Al-phosphate individually or as part of AS04	NLRP3	[6,7,8,9,10] [11,12,13,14,15] ^1^
MF59	Squalene O/W emulsion	NLRP3	[16,17] ^2^ [15,18] ^3^
AS01 AS02	QS-21 + MPL + liposomes QS-21 + MPL + O/W emulsion	NLRP3	[19,20,21] ^4^
AS03	Squalene O/W emulsion + vitamin E	NLRP3	^5^
AS04	MPL adsorbed onto Al(OH)_3_ or AlPO_4_	NLRP3	[6,7,8,9,10] ^6^ [22] ^7^
GLA-SE	GLA + squalene O/W emulsion	NLRP3, Non-canonical	[23]
QS-21	Triterpene saponin fraction purified from *Quillaja saponaria*	NLRP3	[19,20,21]
ISCOM, IMX	Immunostimulating complex containing saponins, phospholipids and cholesterol	NLRP3	[24,25]
CpG-ODN	TLR9 agonist (putative inflammasome-activator)	NLRP3	[26]
Ca_3_(PO_4_)_2_	Calcium phosphate	NLRP3	[27,28]
TDB	Synthetic trehalose-6,6′-dibehenate (an analogue of mycobacterial cord factor trehalose-6,6′-dimycolate)	NLRP3	[29,30]
PLGA	Poly(lactic-co-glycolic acid) microparticles	NLRP3	[31]
Vault NPs	Large cytoplasmic ribonucleoprotein particles	NLRP3	[32]
Chitosan	Biodegradable cationic polymer obtained from chitin	NLRP3	[10,33,34,35,36]
Silica particles	Biocompatible particles	NLRP3	[37,38]
Gold NPs	Gold nanoparticles	NLRP3	[39]
Flagellin	Used as recombinant protein, or encoded in virus replicon or DNA plasmid	NLRC4	[40,41,42,43,44]
DNA vaccines	DNA plasmids or Aim2 encoded in vector as immunopotentiator	AIM2	[45,46,47]
CTB	Cholera toxin B from *Vibrio cholerae*, added to an Ag, or as a DNA vaccine	NLRP3, Pyrin	[48,49,50]
oxPAPC	Oxidized 1-palmitoyl-2-arachidonoyl-sn-glycero-3-phosphorylcholine (generated during tissue damage)	Non-canonical	[51,52,53]

^1^ These studies suggest that the NLRP3 inflammasome is dispensable for the adjuvanticity of Al salts. ^2^ These studies found indications that NLRP3 might be involved in the MoA of MF59 (induction of expression of *caspase-1, IL-1β*, and *IL-1R1*). ^3^ These studies suggest that the NLRP3 inflammasome is dispensable for the adjuvanticity of MF59. ^4^ These studies used components of AS01/AS02: QS-21 + MPL [20]; QS-21 formulated in liposomes [19,21]. ^5^ AS03 is listed as a putative activator of NLRP3, since other squalene O/W emulsions (MF59, GLA-SE) have been suggested as potential inflammasome activators. ^6^ These studies used components of AS04 (Al salts). ^7^ This study showed enhanced adjuvanticity of AS04 compared to Al salts (increased NLRP3 activation through MPL mediated TLR4 priming is a potential explanation). Abbreviations: aluminium (Al), oil-in-water (O/W), adjuvant system (AS), antigen (Ag), glucopyranosyl lipid adjuvant-stable emulsion (GLA-SE), nano particles (NPs), immunostimulating complex (ISCOM), ISCOMATRIX (IMX), poly(lactic-co-glycolic acid) (PLGA), trehalose-6,6′-dibehenate (TDB), cholera toxin B (CTB), oxidized phospholipids (oxPAPC), absent in melanoma 2 (AIM2), nucleotide-binding oligomerization domain, leucine rich repeat and pyrin domain containing protein 3 (NLRP3), NLR family CARD domain-containing protein 4 (NLRC4), CpG oligodeoxynucleotide (CpG-ODN); 3-O-desacyl-4′-monophosphoryl lipid A (MPL); *Quillaja Saponaria* fraction 21 (QS-21).

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
