# Peer review of "Inflammasome-Mediated Immunogenicity of Clinical and Experimental Vaccine Adjuvants"

_vaccines, 2020, doi:10.3390/vaccines8030554_

Round 1
Reviewer 1 Report
This is an interesting review manuscript about perspective for inflammasome-mediated adjuvant properties and potential activities in clinical vaccine fields. Authors have well described that the impact of different inflammasomes on vaccine adjuvant-induced immune response regarding their mechanisms and immunogenicity. In particular, authors highlighted that inflammasomes represent a valuable target of vaccine adjuvants due to induce cellular and humoral arms of adaptive immunity. This review is very informative for vaccine developments for life-threatening diseases including emerging infectious diseases and environmental-related diseases.
Author Response
Response to Reviewer 1 Comments
Comments and Suggestions for Authors
This is an interesting review manuscript about perspective for inflammasome-mediated adjuvant properties and potential activities in clinical vaccine fields. Authors have well described that the impact of different inflammasomes on vaccine adjuvant-induced immune response regarding their mechanisms and immunogenicity. In particular, authors highlighted that inflammasomes represent a valuable target of vaccine adjuvants due to induce cellular and humoral arms of adaptive immunity. This review is very informative for vaccine developments for life-threatening diseases including emerging infectious diseases and environmental-related diseases.
Thank you very much for your kind review.
Reviewer 2 Report
Overall this is a well-written, comprehensive overview of inflammasome-mediated activity of vaccine adjuvants. Some suggestions that could potentially improve the manuscript:
- I think it is important to note that not all clinically approved vaccines have benefited from adjuvants so I would be careful broadly stating that “adjuvants are sophisticated necessary prerequisites”. An example is the licensed quadrivalent meningococcal conjugate vaccines, none of which contain an adjuvant and for which adjuvants (such as aluminium salts) have thus far failed to show a benefit to immune responses.
- When it is stated that “it is important to understand the cellular and molecular mechanisms of action of vaccine adjuvants” it is important to provide a rationale, which presumably is to aid in the design better adjuvant formulations that can result in enhanced pathogen-specific protection.
- It is important to stress the qualitative not just quantitative aspects of adjuvant design/action. Although it is mentioned elsewhere in the manuscript it is missing from the opening paragraph of the Introduction where it should be prominent.
- In that first paragraph in the introduction (or elsewhere), it would also seem important to mention the importance of adjuvants for pediatric immune responses where the immune system is still developing.
- For the comments in lines 230-232, an important caveat, as mentioned above, is that Al-hydroxide has a questionable benefit to MenC conjugate vaccines and is why all three licensed quadrivalent meningococcal conjugate vaccines do not contain aluminium adjuvants.
- It is important to discuss somewhere the difference between the most commonly used preclinical model (mice) and humans when it comes to the action of adjuvants i.e. what applies to mice does not always translate to the clinic.
- For Figure 1, I think CpG oligonucleotides should be included as CpG 1018 adjuvant is included in a licensed product.
- In lines 369-372, it seems to imply that MPL-A is synthetic, which it is not. This needs to be clarified.
Author Response
Response to Reviewer 2 Comments
Comments and Suggestions for Authors
Overall this is a well-written, comprehensive overview of inflammasome-mediated activity of vaccine adjuvants. Some suggestions that could potentially improve the manuscript:
Thank you very much for your kind review.
- I think it is important to note that not all clinically approved vaccines have benefited from adjuvants so I would be careful broadly stating that “adjuvants are sophisticated necessary prerequisites”. An example is the licensed quadrivalent meningococcal conjugate vaccines, none of which contain an adjuvant and for which adjuvants (such as aluminium salts) have thus far failed to show a benefit to immune responses.
Response 1: We agree on this point and changed the first sentence of the abstract in line 16 to say: “In modern vaccines, adjuvants can be sophisticated immunological tools to promote robust and long-lasting protection against prevalent diseases.”
- When it is stated that “it is important to understand the cellular and molecular mechanisms of action of vaccine adjuvants” it is important to provide a rationale, which presumably is to aid in the design better adjuvant formulations that can result in enhanced pathogen-specific protection.
Response 2: We agree on this point and added the following sentence in line 22: Thus, improved understanding of vaccine adjuvant mechanisms may aid in the design of “intelligent” vaccines to provide robust protection from pathogens.
- It is important to stress the qualitative not just quantitative aspects of adjuvant design/action. Although it is mentioned elsewhere in the manuscript it is missing from the opening paragraph of the Introduction where it should be prominent.
Response 3: We agree on this point and have added two sentences in line 48: “Of note, vaccine adjuvant design and choice should always specifically address the targeted pathogen in order to activate the appropriate specific pathways. Thus, adjuvants qualitatively and quantitatively direct the immune system to initiate a pathogen-specific response.“
- In that first paragraph in the introduction (or elsewhere), it would also seem important to mention the importance of adjuvants for pediatric immune responses where the immune system is still developing.
Response 4: We agree on this point and modified the sentence in line 43 to say: “In addition to augmenting the immune response in general, adjuvants can also allow vaccine dose sparing - which would enable to increase global vaccine supply, reduce the number of immunisations, enhance vaccine efficacy in immuno-compromised individuals, such as young children with a developing immune system and the elderly, or broaden the immune response against highly variable pathogens, e.g. influenza”
- For the comments in lines 230-232, an important caveat, as mentioned above, is that Al-hydroxide has a questionable benefit to MenC conjugate vaccines and is why all three licensed quadrivalent meningococcal conjugate vaccines do not contain aluminium adjuvants.
Response 5: We agree on this point and added the following sentence (and ref) in line 265: “Thus, Al-hydroxide has a questionable benefit in meningococcal conjugate vaccines and today licensed quadrivalent meningococcal conjugate vaccines do not contain Al adjuvants [110].”
- It is important to discuss somewhere the difference between the most commonly used preclinical model (mice) and humans when it comes to the action of adjuvants i.e. what applies to mice does not always translate to the clinic.
Response 6: We agree on this point and have added a new sentence (and ref) in line 598: “It must be noted however that, although murine models represent an essential element in the pre-clinical development of novel vaccines, translatability to the clinic is often limited, due to the many differences on the physiological, cellular and molecular level between the species [185].”
- For Figure 1, I think CpG oligonucleotides should be included as CpG 1018 adjuvant is included in a licensed product.
Response 7: We agree on this point and have added CpG-ODN to figure 1.
- In lines 369-372, it seems to imply that MPL-A is synthetic, which it is not. This needs to be clarified.
Response 8: We have removed the word “synthetic” as it is not necessary in the sentence (now line 405).
Reviewer 3 Report
Thank you for the opportunity to review this manuscript. My recommendation is "Minor revision.". The review is well written, and I learned many things from this review. However, a few comments need to be revised before publication as follows.
General comments:
- The abbreviations MoA and SOI are confusing. I recommend not to spell out.
- This manuscript's major weak point is the lack of safety concerns of adjuvants that activate the inflammasome. Please include this.
- It is better to add a Table describing the list of known adjuvant systems that activate inflammasome, resulting in the enhancement of vaccine effects.
Minor comments:
- IL, LPS, ATP, IL-1R1, TNF, and GM-CSF should be spelled out in the first appearances.
- Throughout the manuscript, both in vivo and in vitro should be italic.
Author Response
Response to Reviewer 3 Comments
Comments and Suggestions for Authors
Thank you for the opportunity to review this manuscript. My recommendation is "Minor revision.". The review is well written, and I learned many things from this review. However, a few comments need to be revised before publication as follows.
Thank you very much for your kind review.
General comments:
- The abbreviations MoA and SOI are confusing. I recommend not to spell out.
Response 1: We would like to keep the abbreviation MoA as it is used numerous times in the manuscript. MoA is spelled out in the first appearance in line 53. We removed the abbreviation SOI as it was used two times only throughout the manuscript. We removed “SOI” in line 58 and replaced “SOI” by “site of injection” in line 359.
- This manuscript's major weak point is the lack of safety concerns of adjuvants that activate the inflammasome. Please include this.
Response 2: We agree on this point and have added a new sentence (and ref) in line 601 to address in general the safety of adjuvants that activate the inflammasome: “In this context, it is of paramount importance to thoroughly evaluate safety profiles of new inflammasome activating adjuvants, since inappropriately strong inflammasome activation may induce IL-1β mediated inflammatory symptoms as seen in auto-inflammatory diseases such as Familial Mediterranean fever or cryopyrin-associated periodic syndromes [186].”
- It is better to add a Table describing the list of known adjuvant systems that activate inflammasome, resulting in the enhancement of vaccine effects.
Response 3: We agree on this point and have added a Table in line 66. We also added the sentence “An overview of inflammasome-activating adjuvants can be found in Table 1.” in line 63.
Minor comments:
- IL, LPS, ATP, IL-1R1, TNF, and GM-CSF should be spelled out in the first appearances.
Response 1 (minor comments): We agree on this point and have spelled out IL (line 100), LPS (line 110), ATP (line 114), IL-1R1 (line 172), TNF (line 203), GM-CSF (line 342).
- Throughout the manuscript, both in vivo and in vitro should be italic.
Response 2 (minor comments): We would like to refer this to the Editorial Office, so that the use of italics is in line with the Journal policy.
Reviewer 4 Report
In the present manuscript, Reinke et al. review the role of inflammasomes on vaccine adjuvants. The authors address different aspects of inflammasomes biology and give an extensive overview of the role of inflammasomes in adjuvant properties. They also discussed about current applications of these adjuvants. As a general comment, the paper is well written and clear.
Minor comments:
Line 64 : “are nucleotide-binding domain leucine-rich repeat containing (NLRs) …” are nucleotide-binding domain leucine-rich repeat RECEPTOR (NLRs) containing….
Line 68: “Each inflammasome is activated by a unique ligand”. It is currently well accepted that most inflammasome, such as the NLRP3 inflammasome, have no unique ligand. NLRP3 “senses” several activators or cellular stresses but research did not succeed to find a unique ligand.
Author Response
Response to Reviewer 4 Comments
Comments and Suggestions for Authors
In the present manuscript, Reinke et al. review the role of inflammasomes on vaccine adjuvants. The authors address different aspects of inflammasomes biology and give an extensive overview of the role of inflammasomes in adjuvant properties. They also discussed about current applications of these adjuvants. As a general comment, the paper is well written and clear.
Thank you very much for your kind review.
Minor comments:
Point 1: Line 64: “are nucleotide-binding domain leucine-rich repeat containing (NLRs) …” are nucleotide-binding domain leucine-rich repeat RECEPTOR (NLRs) containing….
Response 1: We agree on this point and have modified the sentence (now line 91) to say: Best characterised PRRs that make up ‘canonical inflammasomes’ are nucleotide-binding domain, leucine-rich repeat receptor (NLR) family, Pyrin domain containing 1 (NLRP1); NLR family, Pyrin domain containing 3 (NLRP3); NLR family, caspase activation and recruitment domain containing 4 (NLRC4); absent in melanoma 2 (AIM2), and pyrin.
Point 2: Line 68: “Each inflammasome is activated by a unique ligand”. It is currently well accepted that most inflammasome, such as the NLRP3 inflammasome, have no unique ligand. NLRP3 “senses” several activators or cellular stresses but research did not succeed to find a unique ligand.
Response 2: We agree on this point and have deleted the sentence (now line 96). We have modified the subsequent sentence (line 93) to say: “In general, inflammasomes recognise diverse pathogen-associated molecular patterns (PAMPs), danger-associated molecular patterns (DAMPs), or the loss of cellular homeostasis.”